# ROBUSTNESS EXPLORATION OF SEMANTIC INFORMATION IN ADVERSARIAL TRAINING

## ABSTRACT

In this paper, we look into the problem of adversarial robustness from the semantic information perspective. We present a novel insight that adversarial attacks destroy the correlation between visual representations and semantic word vectors, and adversarial training fixed it. We further find that the correlation between robust features of different categories is consistent with the correlation between corresponding semantic word vectors. Based on that, we introduce the semantic information to assist model training and propose Semantic Constraint Adversarial Robust Learning (SCARL). Firstly, we follow an information-theoretical lens to formulate the mutual information between the visual representation and the corresponding semantic word vector in the embedding space to bridge the information gap. We further provide a differentiable lower bound to optimize such mutual information efficiently. Secondly, we propose a novel semantic structural constraint, encouraging the trained model to keep the structure of visual representations consistent with that of semantic word vectors. Finally, we combine these two techniques with adversarial training to learn robust visual representation. Experimentally, we conduct extensive experiments on several benchmarks, demonstrating that semantic information is indeed beneficial to model robustness.

## 1 INTRODUCTION

Word embedding is one of the critical technologies in natural language processing (Pennington et al., 2014; Goldberg & Levy, 2014; Tang et al., 2014). It statistics the co-occurrence frequency between pairs of words within a given context in a large-scale training corpus to learn an encoder that can infer vectors for any words in a learned embedded space. A well-trained word embedding model is usually regarded as a knowledge graph (Matthews & Matthews, 2001; Wang et al., 2018), in which the meaning of a word is determined by its relationship to other words in the learned vector space. That is, analogies and correlations between words can be presented by the learned vectors (Hohman et al., 2018; Chersoni et al., 2021), which help the model associate seen objects with unseen objects.

Recently, several works have explored using semantic word/text embedding as supervision signs for zero-shot learning and visual-linguistic pre-training, and have achieved impressive successes in various AI tasks (Qiao et al., 2017; Wang et al., 2018; Radford et al., 2021; Wang et al., 2022). On the other hand, deep neural networks are usually vulnerable to adversarial examples (Szegedy et al., 2014; Goodfellow et al., 2015; Madry et al., 2018; Bhojanapalli et al., 2021), which severely limits their applications in many security scenarios. Fortunately, some studies (Radford et al., 2021; Yu et al., 2022) have shown that the visual model trained with semantic supervised information has much more robust to distribution shift and adversarial examples than standard trained models. As a result, these preliminaries raise a natural question:

*What is the impact of semantic informations on adversarial robustness?*

To answer this question, we explore the relationship between semantic information and model robustness from two aspects: distribution and structural relevance. Firstly, we apply the canonical correlation analysis (CCA) (Hotelling, 1992), which can reflect connections between two random variables. to analyze the distribution relevance between the visual representation and the corresponding semantic word vector. We mainly analyze the correlation coefficient of natural and adversarial image representation with semantic word vector under non-robust and robust models (Madry et al.,

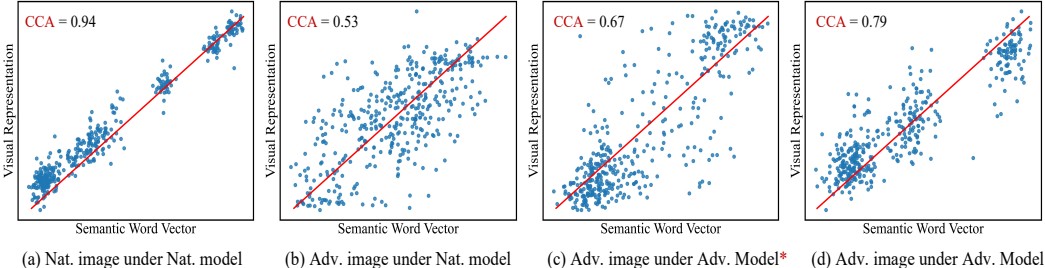

(a) Nat. image under Nat. model    (b) Adv. image under Nat. model    (c) Adv. image under Adv. Model*    (d) Adv. image under Adv. Model

Figure 1: The canonical correlation analysis (CCA) of the natural and adversarial image with the semantic words under the natural and adversarial trained model, respectively. In each plot, we sample 500 image-words pairs to calculate the correlation coefficient. * indicates the model is trained by FGSM, it's less robust than standard adversarial training (d). The larger the CCA, the stronger the correlation between the visual representation and semantic word vector.

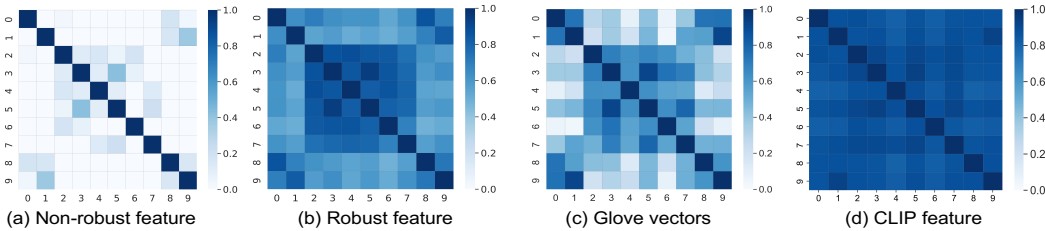

(a) Non-robust feature    (b) Robust feature    (c) Glove vectors    (d) CLIP feature

Figure 2: The similarity matrix between different categories of features learned by different models on CIFAR-10, Different numbers represent different categories. The similarity is calculated by operate inner product between different categories of normalized features. The color is brighter with a larger similarity.

2018). The results in Figure 1 indicate that, for the non-robust model, the representation of the natural image has a high correlation with its corresponding word vector. In contrast, the adversarial image has a lower correlation. This result means that the adversarial attack will destroy the semantic information from the non-robust model, which responds to the previous observations in (Zhang & Zhu, 2019; Ilyas et al., 2019) via a new perspective. For the robust model trained on adversarial examples (Madry et al., 2018), the correlation between the visual representation and word vector has a significant enhancement. As a result, we can summarize a novel intriguing property: *the more robust model, the stronger the correlation*.

Secondly, to verify the semantic word vectors could present the analogies and correlations between words, we visualize the similarity matrix of word vectors generated by a trained Glove (Pennington et al., 2014) on CIFAR-10, which is shown in Figure 2 (c), As can be seen from the figure, the correlation between category 3 (Cat) [1] and category 5 (Dog) is stronger than the correlation between category 3 (Cat) and category 9 (Truck). We further visualize the similarity between different categories of non-robust features, and the similarity of robust features. which are shown in Figure 2 (a) and (b) respectively. We can observe that the robustness feature can also reflect the relatedness between categories, and it is similar to the relatedness reflected by the semantic word vector. However, the non-robust features cannot reflect the association between categories. Recently, CLIP (Radford et al., 2021) uses large-scale image-text pairs to jointly learn semantic representations. Therefore, we also visualize the semantic representation correlation matrix learned by CLIP. which is shown in Figure 2 (d). the semantic representations learned by CLIP present analogies and correlations between categories, but there is a certain gap with the real semantics.

Taking our analysis into consideration, we introduce the semantic information learned by word embedding into model training, which aims at improving the robustness of the current neural networks (He et al., 2016a). We follow an information-theoretical perspective to bridge the information gap

---

[1]The CIFAR-10 contains 10 categories: airplane (0), car (1), bird (2), cat (3), deer (4), dog (5), frog (6), horse (7), ship (8), truck (9).

between visual representations and semantic word vectors, which consists of two key techniques. First, we use mutual information to enhance semantic information in the visual representation, which aims to enhance the correlations via distributional information. Second, we introduce geometric constraints to align the manifold information from the visual representation space to the word vector space, which aims to enhance the correlations via structural information. Finally, we propose the Semantic Constraint Adversarial Robust Learning (SCARL) framework, which combines the above two techniques with adversarial training.

Our contributions are summarized as follows:

- We are the first to explore the correlation between semantic word information and the deep model via the classical CCA method. We find that, the more robust the model, the stronger the correlation between visual representation and semantic word vector.
- We analyze the correlations between different categories of image features, and find that the robust features can reflect the semantic association between categories, which is consistent with the word vector, but the not-robust features can not.
- We introduce a Semantic Constraint Adversarial Robust Learning (SCARL) framework that captures the distributional and structural information from semantic word vectors via mutual information optimization and geometry constraints, to promote robustness.
- We conduct extensive experiments on three widely-used benchmarks. The results show that the proposed SCARL behaves more robust than several state-of-the-art techniques, which demonstrates semantic information indeed helps improve robustness.

## 2 SEMANTIC CONSTRAINT ADVERSARIAL ROBUST LEARNING

### 2.1 PROBLEM SETTING AND NOTATIONS

The robustness against adversarial example attacks has attracted a lot of attention, and various training algorithms have been proposed to mitigate the problem (Madry et al., 2018; Dhillon et al., 2018; Yang et al., 2019; Song et al., 2019; Wu et al., 2020). However, none of these methods could survive the latest gradient masking-based attack (Athalye et al., 2018). In this competition between attackers and defenders, adversarial training (Madry et al., 2018; Zhang et al., 2019) stands out as a promising solution to defend against those strongest adversarial attacks. The approach augments the training data with adversarial inputs produced by an adversarial attack, which can be formulated as a min-max optimization problem:

$$\min_{\theta} \mathbb{E}\left[ \max_{x_{\text{adv}}} \mathcal{L}\Big( F_{\theta}\big(x_{\text{adv}}\big), y \Big) \right], \tag{1}$$

where $F_{\theta}$ is a DNN model with parameters $\theta$, and $\mathcal{L}$ is the loss function of the DNN, $x_{\text{adv}}$ denote adversarial examples.

On the other hand, several works have explored using semantic word embedding as supervision information to improve model performance (Gan et al., 2020; Radford et al., 2021). However, most of them are pre-trained on large-scale image-text datasets and then transferred to downstream tasks. There are several critical limitations of adversarial training: 1) Requesting large-scale image-text datasets, which are collected from the web, often with noisy data; 2) Inconsistency of representation space between visual image and semantic word; 3) There is no reliable robustness evaluation method for multimodal information.

To overcome these limitations, Firstly, we only focus on the essential image classification task, and we use the CIFAR (Krizhevsky et al., 2009) and TinyImageNet (Deng et al., 2009) datasets, which have the one-hot label and the corresponding semantic words. Secondly, for the semantic word vector, we use the Glove (Pennington et al., 2014) or CLIP (Radford et al., 2021) to embed the names or descriptions of the target dataset's classes. Then, we project visual representation and semantic word vector into the consistency manifold, and introduce two constraint strategies from the information and structural perspective to ensure the visual representation manifold is consistent with that of the semantic word vector. Lastly, we combine the proposed semantic constraint techniques with adversarial training to learn robust representation, which is called Semantic Constraint Adversarial Robust Learning (SCARL). The overall framework is shown in Figure 3.

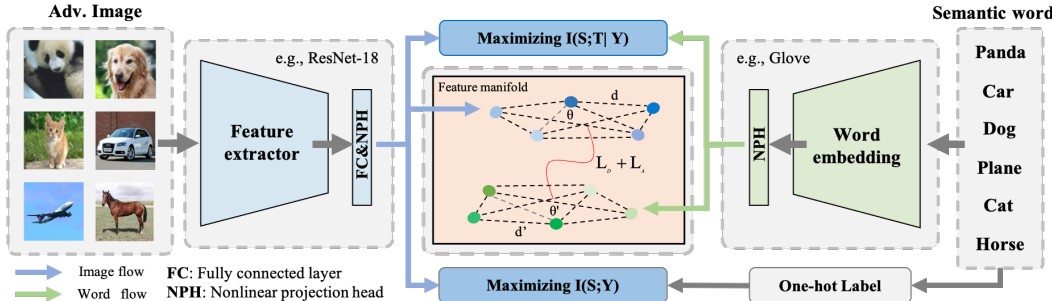

Figure 3: The framework of our SCARL, the visual representation and semantic word vector are extracted by a DNN and a word embedding model respectively. We maximize the mutual information between visual representation and the corresponding semantic word vector by Lemma 2.1, and maximize the mutual information between visual representation and one-hot label by minimizing cross-entropy loss. By geometric structure constraints, we further align the manifold information from the visual representation space to the word vector space. **Best viewed in color.**

**Notations.** We define random variables $X, Y, Z \in p_D(x, y, z)$, where $X$ represents the image input, $Y$ represents the one-hot class label. $Z$ represents the corresponding semantic words. $p_D$ is the data distribution, and $x, y, z$ are the observed values. For the image classification task, our goal is to build a classifier $q(y|F_\theta(x))$, where $F_\theta(x)$ is referred to as our objective model. We define random variable $S = F_\theta(X)$ as the visual representation of $X$ extracted by classifier $F_\theta$, and define random variable $T = E_\theta(Z)$ as word vector of $Z$ encoded by the word embedding model $E_\theta$. Our goal is to train $F_\theta(x)$ such that $X$ is capable of predicting $Y$.

## 2.2 MAXIMIZING LOWER BOUND ON SEMANTIC MUTUAL INFORMATION

In the classification task, the classifier is often trained with cross-entropy loss, which can be viewed as maximizing the mutual information $I(S; Y) = \log \frac{p(S,Y)}{p(S)p(Y)}$, where $p(S, Y)$ denotes the joint probability distribution of $S, Y$, and both $p(S)$ and $p(Y)$ are the marginals. According to variational inference, we can use $q(y|s)$ as a variational distribution of $p(y|s)$, and derive a variational lower bound on $I(S; Y)$ as follows:

$$
\begin{aligned}
I(S; Y) &= H(Y) - \mathbb{E}_{s,y \sim p_D}[-\log q(y \mid s)] + \mathrm{KL}\big(p(\cdot \mid s)\|q(\cdot \mid s)\big) \\
&\geq H(Y) - \mathbb{E}_{s,y \sim p_D}[-\log q(y \mid s)],
\end{aligned}
\tag{2}
$$

where $H(Y)$ is a constant measuring the shannon entropy of $Y$, and $\mathbb{E}_{S,Y}[-\log q(y \mid s)]$ is essentially the cross-entropy loss using $q(y, s)$ for classification. Therefore, the objective of maximizing $I(S; Y)$ can be achieved by minimizing $\mathbb{E}_{S,Y}[-\log q(y \mid s)]$ instead.

However, maximizing $I(S; Y)$ does not take into account the semantic information of the semantic words. To bridge visual representation and semantic word vector. We follow an information-theoretic lens to look into the information gap between visual representation and semantic word vector, and propose maximum semantic mutual information to improve model training. We formulate our objective as follows:

$$
\max I(S; Y, T).
\tag{3}
$$

Our goal is to train the model such that $S$ is capable of predicting $Y$, as well as learning the semantic information from $T$. However, since the objective of equation 3 is difficult to optimize directly, we decompose it into two terms as follows:

$$
I(S; Y, T) = I(S; Y) + I(S; T \mid Y),
\tag{4}
$$

where $I(S; Y)$ measures how well the model can predict the one-hot label, and $I(S; T \mid Y)$ measures how well the visual representation can learn semantic information from the semantic word vector. The $I(S; Y)$ can be optimized using the cross-entropy loss. For optimizing $I(S; T \mid Y)$, inspired by (Hjelm et al., 2018; Tian et al., 2020), we use contrastive learning to derive a lower bound $-\mathcal{L}_{info}$ on the conditional mutual information. To achieve a tractable objective, We introduce the following Lemma:

**Lemma 2.1.** *Given* 1 *congruent pair* $\{x, z, y | C_y = 1\}$ *and* $N$ *incongruent pairs* $\{x_i, z_i, y_i | C_y = 0\}_{i=1}^{N}$, *which is sampled* i.i.d. *from the distribution* $p_D(x, y, z)$, $s = F_\theta(x)$ *and* $t = E_\theta(z)$, $I(S; T \mid Y)$ *is lower bounded by*

$$-\mathcal{L}_{info} = \mathbb{E}_{q(S,T|C_Y=1)}[\log \frac{e^{g_s(s)' g_t(t)/\tau}}{e^{g_s(s)' g_t(t)/\tau} + c}] + N\mathbb{E}_{q(S,T|C_Y=0)}[\log(1 - \frac{e^{g_s(s)' g_t(t)/\tau}}{e^{g_s(s)' g_t(t)/\tau} + c})], \tag{5}$$

where $c$ is the cardinality of the dataset and $\tau$ is a temperature that adjusts the concentration level. $g_s$ and $g_t$ are nonlinear projection heads to transform the representation into the same manifold space and further normalize by the L2-norm. The proof is shown in the supplementary material. By leveraging Lemma 2.1, $\mathcal{L}_{info}$ can be computed using a batch of samples and then minimized for maximizing $I(S; T \mid Y)$.

We further analyse the characteristics of semantic mutual information and its corresponding differential lower bound. From Lemma 2.1, we add $\mathbb{E}_{q(S,T|C_Y=0)}\left[\log\left(1 - \frac{e^{g_s(s)' g_t(t)/\tau}}{e^{g_s(s)' g_t(t)/\tau + c}}\right)\right]$ to relax the bound. Then, our final bound in Lemma 2.1 contains two parts. The first part is to maximize the mutual information between an image and the corresponding semantic words; the second part is to minimize the mutual information between an image and the mismatched semantic words. Intuitively, the formulation of Lemma 2.1 is similar to the fundamental goal for metric learning: *learn a representation that is close in some metric space for "positive" pairs and push apart the representation between "negative" pairs*. Different from traditional metric learning, our approach is based on the perspective of information theory and can be seen as special metric learning by optimizing the mutual information.

### 2.3 SEMANTIC STRUCTURE CONSTRAINT

In the last subsection, we bridged semantic word vector and visual representation through mutual information, but ignored the structure relationship in linguistic words. Based on our findings in Figure 2, the correlation between robust features is similar to that of semantic word vectors. Another reason for using structural constraints is that when a non-robust model is attacked, the manifold of representations is distorted with the disruption of the space structure; thus, maintaining the structure stability is beneficial to the defense. Therefore, we proposed semantic structure constraint loss to keep the structure of images consistent with that of semantic words.

Given a dataset $D$ with $K$ classes. Let $\mathbf{M} \in \mathcal{R}^d$ be a lower-dimensional manifold, where $\{s, t \in M | s = F_\theta(x), t = E_\theta(z), x, z \in D\}$. $s$ was defined using the output of any layer of the network (*e.g.*, a hidden output of the logic layer). We define the visual representation centers as $\mathbf{S}_{image} = \{\mathbf{s}^1, \dots, \mathbf{s}^k | \mathbf{s}^i \in \mathbf{M}\}$, which is the set of $K$ vertices representative of dataset $D$. Each vertex $\mathbf{s}^i$ is the centroid vector representing one class of feature vectors within a neighbourhood region. Similarly, we define the word vector centers $\mathbf{T}_{word} = \{\mathbf{t}^1, \dots, \mathbf{t}^k | \mathbf{t}^i \in \mathbf{M}\}$.

To get $\mathbf{T}_{word}$, we first randomly initialize its value by picking a random position in manifold space. Then, we update $\mathbf{t}^i$ iteratively using the momentum rule:

$$\mathbf{t}_{new}^i = \mathbf{t}^i + m \cdot \left(E_\theta(z|y = i) - \mathbf{t}^i\right), i = 1, \cdots, K, \tag{6}$$

where $\mathbf{t}_{new}^i$ denotes the updated vertex, and the hyperparameter $m \in [0, 1)$ is momentum coefficient. The equation 6 ensures the vertex has a stable step towards the center as training goes on.

For visual representation centers, we construct a new center $\mathbf{s}_{new}^i$ for the observed values of each vertex in each training epoch. $\mathbf{s}_{new}^i$ is estimated by the averaging representation of the same class in the mini-batch samples:

$$\mathbf{s}_{new}^i = \frac{1}{N} \sum_n^N F_\theta(x_n | y_n = i), i = 1, \cdots, K. \tag{7}$$

Then, let the $\mathbf{T}_{word}$ restrict the $\mathbf{S}_{image}$ to make them consistent. To this end, we propose two geometry relation matching metrics: distance-wise and angle-wise. Both of them aim to match the geometry structure information between visual representation and semantic word vector.

**Distance-wise Matching.** For distance-wise metrics, given a pair of representation center $< \mathbf{s}^i, \mathbf{s}^j >$, and a distance-wise function $\phi_D$, we separately calculate the distance (*e.g.*, Euclidean distance) between the two image centers in the representation space:

$$\phi_D(s^i, s^j) = \frac{1}{\mu} \cdot \|\mathbf{s}^i - \mathbf{s}^j\|_2, \tag{8}$$

where $\mu$ is a normalization factor for distance. To focus on the relative distance among other pairs, we set $\mu$ to the average distance between pairs from $\mathbf{S}_{\text{image}}$. $\mu$ is defended as: $\mu = \frac{1}{|\mathbf{S}^2|} \sum_{\mathbf{s}^i, \mathbf{s}^j \in \mathbf{S}} \|\mathbf{s}^i - \mathbf{s}^j\|_2$. Similarly, we can calculate the $\phi_D(\mathbf{t}^i, \mathbf{t}^j)$. Using the distance-wise potential measured in both image and word centers, a distance-wise matching loss is defined as:

$$\mathcal{L}_D = \sum_{\mathbf{s}^i, \mathbf{s}^j \in \mathbf{S}, \mathbf{t}^i, \mathbf{t}^j \in \mathbf{T}} l_\delta \left( \phi_D(\mathbf{s}^i, \mathbf{s}^j), \phi_D(\mathbf{t}^i, \mathbf{t}^i) \right), \tag{9}$$

where $l_\delta$ is smooth L1 loss (Ren et al., 2015), The distance-wise loss matches the relationship of centers by penalizing the distance difference between their output in manifold space.

**Angle-wise Matching.** For angle-wise metrics, given a triplet of training centers $< \mathbf{s}^i, \mathbf{s}^j, \mathbf{s}^k >$, an angle-wise relational potential measures the angle formed by the visual representation and word vector centers in the output manifold space:

$$\phi_A(\mathbf{s}^i, \mathbf{s}^j, \mathbf{s}^k >) = cos\angle ijk = <\mathbf{e}^{ij}, \mathbf{e}^{kj} >, \tag{10}$$

where $\mathbf{e}^{ij} = \frac{\mathbf{s}^i - \mathbf{s}^j}{\|\mathbf{s}^i - \mathbf{s}^j\|_2}, \mathbf{e}^{kj} = \frac{\mathbf{s}^k - \mathbf{s}^j}{\|\mathbf{s}^k - \mathbf{s}^j\|_2}$. Using the angle-wise potentials measured in both visual representation and word vector, an angle-wise matching loss is formulated as:

$$\mathcal{L}_A = \sum_{\mathbf{s}^i, \mathbf{s}^j \mathbf{s}^k \in \mathbf{S}, \mathbf{t}^i, \mathbf{t}^j, \mathbf{t}^k \in \mathbf{T}} l_\delta \left( \phi_A(\mathbf{s}^i, \mathbf{s}^j, \mathbf{s}^k), \ \phi_A(\mathbf{t}^i, \mathbf{t}^j, \mathbf{t}^k) \right). \tag{11}$$

The angle-wise loss matches the structure of the visual manifold with semantic structure by penalizing angular differences.

## 2.4 Adversarial Training with Semantic Constraint

Finally, we combine the proposed semantic constraint techniques with the adversarial training framework to learn robust representation, which is called Semantic Constraint Adversarial Robust Learning (SCARL). Our goal is to maximize equation 4 and maintain the manifold structure under the adversarial setting.

**Maximizing Adversarial $I(S_{\mathbf{adv}}; Y)$:** As mentioned in 2.2, maximizing $I(S_{\text{adv}}; Y)$ can be achieved by minimizing a cross-entropy loss instead. To encourage adversarial robustness, this cross-entropy loss can be upgraded to maximize Kullback-Leibler divergence between natural examples $x_{\text{nat}}$ and adversarial examples $x_{\text{adv}}$ as in (Zhang et al., 2019; Dong et al., 2021). Therefore, the objective function can be formulated as follows:

$$\mathcal{L}_{\text{adv}} = \mathcal{L}_{\text{ce}}(F_\theta(x_{\text{adv}}), y) + \beta \cdot \mathcal{KL}\left(P(\cdot|x_{\text{adv}}) \| P(\cdot|x_{\text{nat}})\right). \tag{12}$$

**Maximizing Adversarial $I(S_{\mathbf{adv}}; T \mid Y)$:** To maximize the semantic mutual information $I(S_{\text{adv}}; T \mid Y)$. According to Lemma 2.1 , we formulate the objective to minimize $\mathcal{L}_{info}$ as follows:

$$\begin{aligned}
\mathcal{L}_{info} = \ &\mathbb{E}_{q(S,T|C_Y=1)}\left[-\log \frac{e^{g_s(s)' g_t(t)/\tau}}{e^{g_s(s)' g_t(t)/\tau} + c}\right] \\
&+ N\mathbb{E}_{q(S,T|C_Y=0)}\left[-\log(1 - \frac{e^{g_s(s)' g_t(t)/\tau}}{e^{g_s(s)' g_t(t)/\tau} + c})\right],
\end{aligned} \tag{13}$$

where $s_{adv} = F_\theta(x_{adv}), t = E_\theta(z_y)$, and $c$ is cardinality of the dataset and $\tau$ is a temperature that adjusts the concentration level. $g_s$ and $g_t$ are nonlinear projection heads.

**Restricting $\mathbf{S}_{\mathbf{image}}$ with $\mathbf{T}_{\mathbf{word}}$:** Based on the geometric matching, we formulate the semantic structure constraint loss as follows:

$$\mathcal{L}_{\text{struc}} = \mathcal{L}_{\text{D}} + \mathcal{L}_A \tag{14}$$

Table 1: Robustness accuracy comparison of the proposed approach and baseline models under different attack methods under the $\ell_\infty$ norm with $\epsilon = 8/255$ on different datasets. All the models are based on pre-activation ResNet-18 architecture. We choose the best checkpoint according to the highest robust accuracy on the test set under PGD-10. The best results are **blodfaced**.

| Dataset | Method | Natural | FGSM | PGD-100 | CW-100 | Square | AutoAttack |
|---------|--------|---------|------|---------|--------|--------|------------|
| CIFAR-10 | AT | 83.14 | 57.98 | 51.11 | 49.83 | 54.42 | 47.56 |
| | TLA | 83.75 | 58.17 | 51.35 | 50.54 | 56.17 | 48.17 |
| | ACL | **84.03** | 58.48 | 52.75 | 50.78 | **56.43** | 48.86 |
| | TRADES | 82.76 | **58.85** | 53.43 | 50.91 | 54.85 | 49.34 |
| | **SCARL** | 80.57 | 58.44 | **54.22** | **51.23** | 55.93 | **50.42** |
| CIFAR-100 | AT | 57.96 | 32.64 | 29.29 | **27.46** | 28.26 | 24.06 |
| | TLA | 56.51 | 32.59 | 29.16 | 27.44 | 30.57 | 25.15 |
| | ACL | 57.63 | 32.58 | 29.01 | 27.53 | 30.93 | 25.11 |
| | TRADES | **58.95** | 33.19 | 30.07 | 26.53 | 30.43 | 25.33 |
| | **SCARL** | 58.74 | **34.13** | **31.61** | 27.41 | **31.13** | **26.32** |
| Tiny-ImageNet | AT | 48.49 | 25.37 | 23.07 | 20.72 | 26.42 | 18.59 |
| | TLA | 47.36 | 25.31 | 23.09 | 21.06 | 27.87 | 18.84 |
| | ACL | 48.04 | 25.06 | 22.79 | 20.82 | 26.01 | 18.62 |
| | TRADES | **49.89** | 25.32 | 23.18 | 19.59 | 25.41 | 18.47 |
| | **SCARL** | 49.35 | **26.10** | **23.51** | **21.67** | **27.07** | **19.29** |

**Overall Objective Loss:** We integrate all the above technologies into an end-to-end training framework. The overall loss function of our algorithm is formulated as follows:

$$\mathcal{L}_{obj} = \mathcal{L}_{adv} + \lambda_1 \cdot \mathcal{L}_{info} + \lambda_2 \cdot \mathcal{L}_{\text{struc}}, \tag{15}$$

where $\lambda_1$ and $\lambda_2$ are hyperparameters to control the relative importance among the three losses. The training process not only maximizes the mutual information between visual representation and semantic word vector in the consistent manifold space, and also captures the semantic structure information, which enables our model to learn more semantic and robust representations that are not sensitive to the input perturbation.

## 3 EXPERIMENTS

### 3.1 EXPERIMENTAL SETUP

**Datasets.** We compare the proposed methods with the baselines on widely-used benchmark datasets, including: CIFAR-10, CIFAR-100 (Krizhevsky et al., 2009) and TinyImageNet (Deng et al., 2009). These datasets can easily obtain one-hot labels and the corresponding semantic words.

**Baselines Setup.** We compare the robustness of our proposed SCARL with some classical adversarial training methods, including standard AT(Madry et al., 2018), TRADES (Zhang et al., 2019). triplet loss adversarial training (TLA) (Mao et al., 2019) and adversarial training with contrastive learning (ACL) (Jiang et al., 2020). We test the defence under different white- and black-box attacks including FGSM (Goodfellow et al., 2015), PGD (Madry et al., 2018), CW (Carlini & Wagner, 2017), Square (Andriushchenko et al., 2019) and AutoAttack (Croce & Hein, 2020).

**Model Details.** We adopt a pre-activation ResNet-18 He et al. (2016b) as the image feature extractor, then follow a nonlinear projection with one additional hidden layer (and ReLU activation). The output vector (64-D) is normalized by its L2-norm. This is the representation of the image. For the word embedding model, we use the Glove (Pennington et al., 2014), which is trained with Wikipedia (Scheepers, 2017) and Gigaword (Rush et al., 2015) datasets. The word vector is also projected into a vector (64-D) by a nonlinear projection head and normalized. This is the representation of semantic words.

**Training Details.** For training, the initial learning rate is $\gamma = 0.1$, and the learning rate schedule is [0.1, 0.01, 0.001] for all datasets, the decay epoch schedule is [75, 90] for CIFAR and [50, 55] for

TinyImageNet. The training scheduling of 100 epochs for CIFAR and 60 epochs for TinyImageNet. Following (Rice et al., 2020), we adopt the common setting that the threat model with radius 8/255, with the PGD attack taking 10 steps of size 2/255 on all datasets. We train with the SGD optimizer with momentum 0.9, weight decay $5 \cdot 10^{-4}$ using a batch size of 128. We performed standard data augmentation including random crops and random horizontal flips during training. For hyperparameter, we set the hyperparameter $\lambda_1 = 1.0$ and $\lambda_2 = 1.0$ along a Gaussian ramp curve in equation 15, the $\beta = 6$ in equation 12. Our implementation is based on PyTorch and the code to reproduce our results will be available.

## 3.2 MAIN RESULTS

To verify the impact of the semantic information on model robustness. we train a natural model and robust model with semantic information (SemInfo), and compare it with another model trained without semantic information. Table 2 shows the results. We find that the model using semantic information is more robust against simple attacks such as FGSM, but still not robust against stronger attacks. Further, we combine semantic information with standard adversarial training (Madry et al., 2018), The Seminfo improves the robustness by $1.25\%$ under standard AutoAttack, which demonstrated the semantic information is beneficial to the model's robustness.

Table 2: The robustness results in CIFAR-10 under the $\ell_\infty$ norm with $\epsilon = 8/255$.

| Method | FGSM | AutoAttack |
|---|---|---|
| Nat. training | 33.24 | 0.0 |
| Nat. training + SemInfo | 36.32 | 0.0 |
| Adv. training | 57.98 | 47.56 |
| Adv. training + SemInfo | 58.63 | 48.81 |

In order to further verify the influence of semantic information on the robustness of the model, we calculate the CCA of our SCARL where the model trained with the equation 15, the CCA is 0.84 [2], which means visual representations learned by our SCARL are more semantic information than standard adversarial training (Madry et al., 2018). We further report the results under different white- and black-box attacks at the best checkpoint, which is selected based on the performance under the PGD-10 attack. The results are shown in Table 1. The proposed method SCARL achieves the best robustness against the strongest attacks AutoAttack on both CIFAR and TinyImageNet, where every small margin of improvement is significant.

We further verify the robustness of our method under adaptive attack (Akhtar & Mian, 2018) where the attacker has the knowledge and white-box access to the word embedding model. We use the PGD-based adaptive attack to evaluate AT, TRADES and our SCARL model under ResNet-18 on CIFAR10. The results are shown in Table 3 and demonstrate our SCARL can still defend against the adaptive attack.

Table 3: Robustness comparison of different adversarially trained models under adaptive attacks.

| Model | Natural | PGD-CE | PGD-KL | PGD-Info |
|---|---|---|---|---|
| AT | **83.75** | 51.35 | 51.88 | 52.36 |
| TRADES | 82.76 | 52.73 | 52.33 | 53.97 |
| SCARL | 80.57 | **54.22** | **53.81** | **54.75** |

We also tested the performance under WideResNet (Zagoruyko & Komodakis, 2016), the results are shown in supplementary material since space limitations. In addition, We plot the results of testing accuracy over epochs and evaluate adversarial accuracy against PGD attacks under different attack budgets with a fixed attack step of 10, and we also conduct experiments using PGD attacks with different attack iterations with a fixed attack budget of $8/255$. The results shown in Figure 4 (a-c). Our SCARL is better than standard AT and TRADES at a larger budget, besides, our SCARL is stable against large iterations attacks, *e.g.*, PGD attack with 500 step iterations. Therefore, the results demonstrate the effectiveness of our proposed SCARL.

## 3.3 ABLATION STUDIES

For ablation studies, all comparative experiments were performed on the CIFAR-10, and all other hyper-parameters were kept exactly the same other than the contrast variable used.

**Ablation Different Techniques.** We validate the two proposed techniques in SCARL, of which results are given in Table 4, From Table 4, we can observe significant performance gains by each technique. This confirms the merits of our proposed techniques.

---

[2]Due to space limitations, we show the visualization of CCA in the supplementary material.

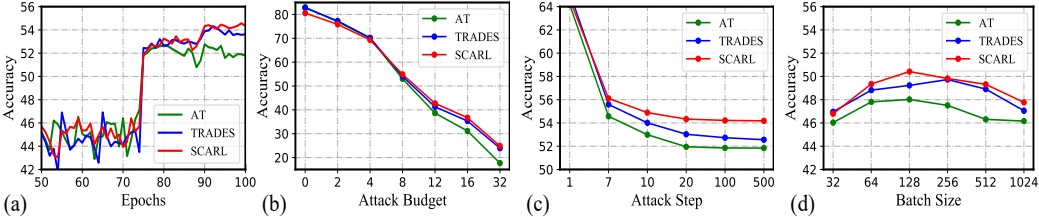

(a)  (b) Attack Budget  (c) Attack Step  (d) Batch Size

Figure 4: (a) is the test accuracy curves (under PGD-10). (b) and (c) is the test accuracy under PGD attack with different attack budgets and attack iterations, respectively. (d) is the test accuracy under different batch sizes. All these experiments were conducted on CIFAR-10.

Table 4: Impact of different techniques of our proposed method. **Base** indicates the model trained with equation 12.

| Method | Natural | AutoAttack |
|---|---|---|
| Base | 80.44 | 49.36 |
| Base + $\mathcal{L}_{info}$ | 81.38 | 50.29 |
| Base + $\mathcal{L}_{struc}$ | 80.96 | 50.08 |
| Base + $\mathcal{L}_{info}$ + $\mathcal{L}_{struc}$ | 80.57 | **50.42** |
| Base + $\mathcal{L}_{infoNCE}$ | **81.68** | 49.95 |
| Base + $\mathcal{L}_{infoNCE}$ + $\mathcal{L}_{struc}$ | 81.31 | 50.13 |

Table 5: The effect of different word embedding. **Base** indicates the model trained with equation 12.

| Method | Natural | AutoAttack |
|---|---|---|
| Base | 80.44 | 49.36 |
| Random Embedding | 79.69 | 48.87 |
| NN Embedding | **81.23** | 49.38 |
| CLIP Embedding | 80.38 | 50.04 |
| Glove Embedding | 80.57 | **50.42** |

**Comparison with InfoNCE.** The proposed $\mathcal{L}_{info}$ is similar to InfoNCE (Hjelm et al., 2018). InfoNCE is an alternative contrastive objective that selects a single positive out from a set of distractors via a softmax function. We compare InfoNCE with our $\mathcal{L}_{info}$ when using the same number of negatives. The Table 4 show that our $\mathcal{L}_{info}$ outperforms InfoNCE. This confirms the merits of the proposed $\mathcal{L}_{info}$.

**Effect of Semantic Embeddings.** We use different word embedding to verify semantic information is beneficial to robustness. we design four embedding schemes: a) **Random**: a random vector as the semantic word vector; b) **NN**: word vector is generated by a learnable neural network embedded layer; c) **CLIP**: word vector is generated by a trained CLIP model; d) **Glove**: word vector is generated by a trained Glove model. The results are shown in Table 5. It is observed that the stronger the semantic information, the higher the robustness.

**Effect of Batch Size.** Theoretically, the proposed $\mathcal{L}_{info}$ and InfoNCE can benefit from a large batch size. To evaluate the effect of batch sizes, we test six values of batch size and show the results in Figure 4(d). For the standard adversarial training, as the mini-batch sizes get larger, the performance drops dramatically. This proves that adversarial training is not suitable for large batch sizes on CIFAR-10. The other reason for the $\mathcal{L}_{info}$ is not to benefit from large batch size is that, under a given dataset, the semantic words of the data category are fixed, when calculating $\mathcal{L}_{info}$, even if using the large batch size, the negative samples in $\mathcal{L}_{info}$ is still restricted by the number of semantic words.

## 4 CONLUSION

In this paper, we analyzed the relationship between semantic information and model robustness from a distribution and structural perspective. which shows the robustness of image representation is closely related to semantic information. Based on our findings, we proposed Semantic Constraint Adversarial Robustness Learning (SCARL), which learns visual representation by capturing semantic word distribution and structure information. Experimentally, we demonstrated the effectiveness of the proposed SCARL in multiple benchmark datasets and revealed that semantic information could indeed improve model robustness.

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

## A THE PROOF OF LEMMA 2.1

**Lemma 2.1.** Given 1 congruent pair $\{x, z, y | C_y = 1\}$ and $N$ incongruent pairs $\{x_i, z_i, y_i | C_y = 0\}_{i=1}^{N}$, which is sampled *i.i.d.* from the distribution $p_D(x, y, z)$, $s = F_\theta(x)$ and $t = E_\theta(z)$, $I(S; T \mid Y)$ is lower bounded by

$$-\mathcal{L}_{info} = \mathbb{E}_{q(S,T|C_Y=1)}[\log \frac{e^{g_s(s)'g_t(t)/\tau}}{e^{g_s(s)'g_t(t)/\tau} + c}] + N\mathbb{E}_{q(S,T|C_Y=0)}[\log(1 - \frac{e^{g_s(s)'g_t(t)/\tau}}{e^{g_s(s)'g_t(t)/\tau} + c})], \tag{16}$$

where $c$ is the cardinality of the dataset and $\tau$ is a temperature that adjusts the concentration level. $g_s$ and $g_t$ are nonlinear projection heads to transform the representation into the same manifold space and further normalize by L2-norm.

*Proof.* For deriving the lower bound, we define the mutual information containing the joint distribution $p(S, T)$ and the product of marginal distributions $p(S)p(T)$. Therefore, we can maximize the mutual information between visual representation and word vector via optimizing KL-divergence between these distributions. To this end, we define a distribution $q$ with the latent variable $C_Y$. When $C_Y = 1$, the pair $\langle S, T \rangle$ have the same one-hot label, which is drawn from the joint distribution. On the contrary, when $S$ and $T$ have different one-hot labels ($C_Y = 0$), the pair is independent of each other, which is drawn from the product of marginal. As a result, the formulations can be written as:

$$q(S, T | C_Y = 1) = p(S, T | Y),$$
$$q(S, T | C_Y = 0) = p(S | Y)p(T | Y). \tag{17}$$

Suppose that, there are 1 congruent pair, which is drawn from the joint distribution, and $N$ incongruent pairs, which are drawn from the product of marginal. We can define the priors on the latent $C$ are:

$$q(C_Y = 1) = \frac{1}{1 + N}, \ q(C_Y = 0) = \frac{N}{1 + N}. \tag{18}$$

By using Bayes' rule, the posterior for class $C_Y$=1 is:

$$q(C_Y = 1 | S, T) = \frac{q(S, T | C_Y = 1)q(C_Y = 1)}{q(S, T | C_Y = 0)q(C_Y = 0) + q(S, T | C_Y = 1)q(C_Y = 1)}$$
$$= \frac{p(S, T | Y)}{p(S, T | Y) + Np(S | Y)p(T | Y)}. \tag{19}$$

Then, we bridge a connection with mutual information as:

$$\log q(C_Y = 1 | S, T) = \log \frac{p(S, T | Y)}{p(S, T | Y) + Np(S | Y)p(T | Y)}$$
$$= -\log \left(1 + N\frac{p(S | Y)p(T | Y)}{p(S, T | Y)}\right) \tag{20}$$
$$\leq -\log N + \log \frac{p(S, T | Y)}{p(S | Y)p(T | Y)}.$$

Taking expectation on both sides *w.r.t.* $p(S, T | Y)$ and rearranging, we have:

$$I(S; T | Y) \geq \log N + \mathbb{E}_{q(S,T|C_Y=1)} \log q(C_Y = 1 | S, T), \tag{21}$$

where $I(S; T | Y)$ is the conditional mutual information between the distribution of visual representations and word vectors under the same one-hot label. Thus, maximizing $\mathbb{E}_{q(S,T|C_Y=1)} \log q(C_Y = 1 | S, T)$ *w.r.t.* the parameters of the objective model is to increase a lower bound on mutual information. However, it is intractable to directly optimize such a lower bound since we do not know the true distribution $q(C_Y = 1 | S, T)$. To achieve a tractable objective, We introduce the following Lemma:

In equation 21, maximizing $\mathbb{E}_{q(S,T|C_Y=1)} \log q(C_Y = 1 | S, T)$ *w.r.t.* the parameters of the objective model is to increase a lower bound on mutual information. However, it is intractable to directly optimize such a lower bound since we do not know the true distribution $q(C_Y = 1 | S, T)$. Thus, we estimate it by fitting a model $h : \{S, T\}-> [0, 1]$ to samples from the data distribution

$q(C_Y = 1|S, T)$. We maximize the log-likelihood of the data under this model (a binary classification problem):

$$\mathcal{L}_{est}(h) = \mathbb{E}_{q(S,T|C_Y=1)}\left[\log h(S, T)\right] + N\mathbb{E}_{q(S,T|C_Y=0)}\left[\log\left(1 - h(S, T)\right)\right]. \tag{22}$$

$$h^\star = \arg\max_h \mathcal{L}_{est}(h). \tag{23}$$

According to Gibbs' inequality, we have:

$$q(C_Y = 1|S, T) = h^\star = \arg\max_h \mathcal{L}_{est}(h). \tag{24}$$

The details of equation 24 can be found in Appendix B. Thus, we can rewrite equation 21 in terms of $h^\star$:

$$I(S; T|Y) \geq \log N + \mathbb{E}_{q(S,T|C_Y=1)}[\log h^\star], \tag{25}$$

The optimal $h^\star$ is an estimator whose expectation lower-bounds mutual information. Our goal is to learn a model $F_\theta$ that maximizes the mutual information between visual representations and corresponding semantic word vectors. As a result, we have the following optimization problem:

$$F_\theta = \arg\max_\theta \mathbb{E}_{q(S,T|C_Y=1)}\left[\log h^\star(S, T)\right]. \tag{26}$$

However, this is still difficult to optimize, since the estimator $h^\star$ depends on the current model $F_\theta$. To hand this problem, we further relax the bound in equation 25 to:

$$\begin{aligned} I(S; T|Y) &\geq \log N + \mathbb{E}_{q(S,T|C_Y=1)}\left[\log h^\star(S, T)\right] + N\mathbb{E}_{q(S,T|C_Y=0)}\left[\log\left(1 - h^\star(S, T)\right)\right] \\ &= \log N + \mathcal{L}_{est}(h^\star) = \log N + \max_h \mathcal{L}_{est}(h) \\ &\geq \log N + \mathcal{L}_{est}(h). \end{aligned} \tag{27}$$

Since the $N\mathbb{E}_{S,T|C_Y=0}\left[\log\left(1 - h^\star(S, T)\right)\right]$ is strictly negative, the inequality still holds by adding such term into equation 26. Then, optimizing equation 27 *w.r.t.* the $F_\theta$ can be reformulated as follows:

$$\begin{aligned} F_\theta &= \arg\max_\theta \max_h \mathcal{L}_{est}(h) \\ &= \arg\max_\theta \max_h \left(\mathbb{E}_{q(S,T|C_Y=1)}\left[\log h(S, T)\right] + N\mathbb{E}_{q(S,T|C_Y=0)}\left[\log\left(1 - h(S, T)\right)\right]\right). \end{aligned} \tag{28}$$

At last, equation 28 is our final learning objective, which jointly optimizes $F_\theta$ together with learning $h$. Note that, due to equation 27, $F_\theta = \arg\max_\theta \max_h \mathcal{L}_{est}(h)$, for any $h$, is always the lower bound on the mutual information, which means our objective equation 28 does not depend on $h$ being optimized perfectly. Therefore, we define $h$ with any family of functions that satisfy $h : \{S, T\} \rightarrow [0, 1]$. In practice, we define the $h$ as follows:

$$h(S, T) = \frac{e^{g_s(s)'g_t(t)/\tau}}{e^{g_s(s)'g_t(t)/\tau} + c} \tag{29}$$

where $c$ is the cardinality of the dataset and $\tau$ is a temperature that adjusts the concentration level. $g_s$ and $g_t$ are nonlinear projection heads to transform the representation into the same manifold space and further normalize by L2-norm. Therefore, $I(S; T \mid Y)$ is lower bounded by

$$-\mathcal{L}_{info} = \mathbb{E}_{q(S,T|C_Y=1)}[\log\frac{e^{g_s(s)'g_t(t)/\tau}}{e^{g_s(s)'g_t(t)/\tau} + c}] + N\mathbb{E}_{q(S,T|C_Y=0)}[\log(1 - \frac{e^{g_s(s)'g_t(t)/\tau}}{e^{g_s(s)'g_t(t)/\tau} + c})], \tag{30}$$

$\square$

## B   THE PROOF OF EQUATION 24

*Proof.* In equation 24, the first item $q(C = 1|S, T)$ presents the true distribution of the data that is from the same one-hot label, where $C$ is a binary variable to judge whether the label is correct. Therefore, it is intuitive that the $q(C|S, T)$ can be modeled as a Bernoulli distribution, such as $h : (S, T) \rightarrow [0, 1]$. For convenience, we define $h'(S, T, C = 1) = h(S, T)$ and $h'(S, T, C = 0) = 1 - h(S, T)$, then the log-likelihood is:

$$\mathbb{E}_{c \in q(C|S,T)}\left[\log h'(S, T, C = c)\right]. \tag{31}$$

By using Gibbs' inequality, the max likelihood fit is $h'(S, T, C = c) = q(C = c|S, T)$, which also implies that $h(S, T) = q(C = 1|S, T)$.

Then, we rewrite our objective in equation 22 as follows:

$$\mathbb{E}_{s,t \in q(S,T)} \left[ \mathbb{E}_{c \in q(C|S=s,T=t)} \left[ \log h'(S = s, T = t, C = c) \right] \right] \tag{32}$$

$$= \mathbb{E}_{c,s,t \in q(C,S,T)} \left[ \log h'(S = s, T = t, C = c) \right] \tag{33}$$

$$= \mathbb{E}_{s,t \in q(S,T|C=1)q(C=1)} \left[ \log h(S = s, T = t) \right] + \mathbb{E}_{s,t \in q(S,T|C=0)q(C=0)} \left[ \log 1 - h(S = s, T = t) \right] \tag{34}$$

$$= \frac{1}{N+1} \mathbb{E}_{s,t \in q(S,T|C=1)} [\log h(S = s, T = t] + \frac{N}{N+1} \mathbb{E}_{s,t \in q(S,T|C=0)} [\log 1 - h(S = s, T = t] \tag{35}$$

Notice that equation 35 is proportional to equation 22 from the Appendix A. For sufficiently expressive $h$, then, each term inside the expectation in equation 32 can be maximized, resulting in $h^\star(S = s, T = t) = q(C = 1|S = s, T = t)$ for all $s$ and $t$. $\qquad\square$

## C   PERFORMANCE UNDER WIDERESNET

Many works have demonstrated larger model capacity can usually lead to better adversarial robustness (Madry et al., 2018; Gowal et al., 2020; Pang et al., 2021). Therefore, we employ the large-capacity network, *e.g.,*, Wide ResNet(Zagoruyko & Komodakis, 2016). Table 6 reports the best test robustness against AA on the CIFAR-10. We compare several state-of-the-art adversarial trained models on robust benchmark (Croce et al., 2020). our SCARL achieves 54.65% and 56.42% with AWP respectively, which makes the trained model surpass the previously state-of-the-art models reported by the benchmark. where every small margin of improvement is significant. **Notes**, our experiments did not use additional datasets.

Table 6: Robustness accuracy comparison of the proposed approach and several state-of-the-art models under AA at $\ell_\infty$ norm with $\epsilon = 8/255$ on CIFAR-10. Most of the results are directly copied from the leaderboards (Croce & Hein, 2020). $\star$ indicates the model is re-produced by ourselves.

| Method | Architecture | Nat. | AA |
|---|---|---|---|
| TRADES (Zhang et al., 2019)(ICML2019) | WideResNet-34-10 | 84.92 | 53.08 |
| Overfitting (Rice et al., 2020)(ICML2020) $\star$ | WideResNet-34-10 | 85.18 | 53.14 |
| SAT (Huang et al., 2020)(NeurIPS2020) | WideResNet-34-10 | 83.48 | 53.34 |
| Overfitting (Rice et al., 2020)(ICML2020) | WideResNet-34-20 | 85.35 | 53.42 |
| FAT (Zhang et al., 2020)(ICML2020) | WideResNet-34-10 | 84.52 | 53.51 |
| LBGAT (Cui et al., 2021)(ICCV2021) | WideResNet-34-20 | **88.70** | 53.57 |
| HE (Pang et al., 2020)(NeurIPS2020) | WideResNet-34-20 | 85.14 | 53.74 |
| Bag of Tricks (Pang et al., 2021)(ICLR2021) $\star$ | WideResNet-34-10 | 86.28 | 53.84 |
| LAS-AT Jia et al. (2022)(CVPR2022) | WideResNet-34-10 | 85.24 | 54.13 |
| Bag of Tricks (Pang et al., 2021)(ICLR2021) | WideResNet-34-20 | 86.43 | 54.39 |
| SCARL (Ours) | WideResNet-34-10 | 84.33 | 54.65 |
| TRADES + AWP (Wu et al., 2020)(NeurIPS2020) | WideResNet-34-10 | 85.26 | 56.17 |
| LAS-AT + AWP Jia et al. (2022)(CVPR2022) | WideResNet-34-10 | 84.98 | 56.26 |
| SCARL + AWP | WideResNet-34-10 | 84.70 | **56.42** |

# D   VISUALIZATION OF CCA

We calculate the CCA of our SCARL, which is shown in Figure 5. It is observed that the visual representations learned by our SCARL are more semantic information than standard adversarial training (AT) Madry et al. (2018).

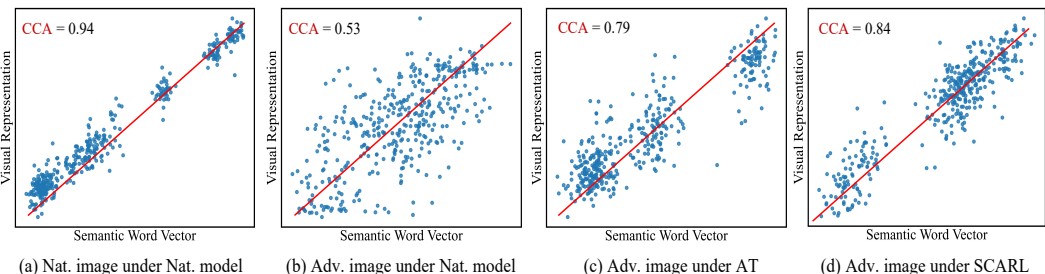

(a) Nat. image under Nat. model (b) Adv. image under Nat. model (c) Adv. image under AT (d) Adv. image under SCARL

Figure 5: The canonical correlation analysis (CCA) of the natural and adversarial image with the semantic words under the non-robust and robust model, respectively. In each plot, we sample 500 image-words pairs to calculate the correlation coefficient. The larger the CCA, the stronger the correlation between the visual representation and semantic word vector.

# E   RELATED WORK

The problem of adversarial examples was first studied in (Szegedy et al., 2014). Then, many works proposed a series of adversarial attack methods (Moosavi-Dezfooli et al., 2016; Papernot et al., 2016a; Carlini & Wagner, 2017; Croce & Hein, 2020), which puts severe limitations on the application of deep learning in security-critical scenarios. With the rapid development of attack methods, considerable efforts have been devoted to defending against adversarial attacks, such as defensive distillation (Papernot et al., 2016b), manifold-projection (Samangouei et al., 2018), pre-processing (Guo et al., 2018; Yang et al., 2019), verification and provable defences (Raghunathan et al., 2018; Salman et al., 2019), and Adversarial Training (Madry et al., 2018; Zhang et al., 2019). Among them, adversarial training has been demonstrated to be a practical approach for strengthening the robustness of deep neural networks (Athalye et al., 2018). Adversarial training involves the min-max optimization problem as Eq. equation 1. The inner maximization can be solved approximately, using FGSM or PGD attack. The outer minimization can be achieved by minimizing cross-entropy loss instead. Based on that, a number of new adversarial training methods have also been devoted from different aspects including designing new adversarial regularization (Zhang et al., 2019; Mao et al., 2019), robustness architecture search (Guo et al., 2020; Hosseini et al., 2021), training strategy (Wong et al., 2020; Pang et al., 2021) and data augmentation (Carmon et al., 2019; Rebuffi et al., 2021). To the best of our knowledge, we are the first to explore the impact of semantic information for adversarial training.

# F  MORE ABLATION RESUTLS

## F.1  COMPARISON WITH TRADES WITH DIFFERENT SETTINGS

we combined the semantic information (SemInfo) with TRADES, the balance parameter $\beta$ set as 1 and 6. The results are shown above. We can see the robustness can be further improved.

Table 7: Robustness accuracy comparison

| Method | FGSM | AutoAttack |
|---|---|---|
| Adv. training | 83.14 | 47.56 |
| Adv. training + SemInfo | 82.72 | 48.81 |
| TRADES ($\beta = 1$) | **87.49** | 44.80 |
| TRADES ($\beta = 1$) + SemInfo | 86.88 | 45.21 |
| TRADES ($\beta = 6$) | 82.76 | 49.34 |
| TRADES ($\beta = 6$) + SemInfo | 82.22 | **49.72** |

## F.2  IMPACT OF THE WORD EMBEDDING MODELS

We provide the results obtained by training several different word embedding models, as shown in the following table: We can see that different word embedding models can improve the robustness of the model.

Table 8: Different Glove model

| Method | Natural | AutoAttack |
|---|---|---|
| Random Embedding | 79.69 | 48.87 |
| CLIP | 80.38 | 50.04 |
| Glove (Twitter) | 80.71 | 50.12 |
| Glove (Common Craw) | 81.06 | 50.25 |
| Glove (Wikipedia 2014 and Gigaword) | 80.57 | **50.42** |

