# OpenReview forum: "Robustness Exploration of Semantic Information in Adversarial Training"
_ICLR.cc/2023/Conference — Submitted to ICLR 2023_

### Official Review · Reviewer_ARcW · 2022-10-25

**Confidence:** 3
**Correctness:** 3
**Technical Novelty And Significance:** 3
**Empirical Novelty And Significance:** 2
**Recommendation:** 5

**Clarity, Quality, Novelty And Reproducibility:**

Clarity: The paper is well-written and easy to follow.

Quality & Novelty: The paper seems novel and has some potential. But the empirical results are not convincing enough to show significant advantages over existing works.

Reproducibility: The paper contains enough information to reproduce the method.

**Strength And Weaknesses:**

Strengths:
1. The paper is well organized. The writing is clear, and it is very easy to follow.
2. The idea of maintaining the semantic information of the class labels during adversarial training is sound.
3. The proposed method is evaluated on benchmark datasets with various adversarial attacks. The performance is generally competitive compared to prior arts. It also conducts extensive ablative studies on the components of the method.
4. The paper provides clear details on reproducing the method.

Weakness:
1. The improvements over previous methods are not convincing enough. The margins are small, especially considering that the proposed method requires more information, i.e., a pretrained word embedding. This might require more resources in real-world applications than other methods, making it less attractive.

**Summary Of The Paper:**

This paper proposes a new adversarial training scheme that leverages the semantic information from language embeddings to improve the robustness of the model to adversarial attacks. The proposed method is evaluated against various adversarial attacks and compared to multiple methods, showing competitive results.

**Summary Of The Review:**

This paper proposed a novel adversarial training method that enforces the alignment between the visual feature and the label semantics from a pretrained word embedding to improve the robustness of adversarial attacks. It shows some potential, but I am not very convinced by the results, considering it requires much more resources than other methods that do not rely on external information. Therefore I am more inclined to rejection.

---

> ### Author Response · Authors · 2022-11-16
> **Response to Reviewer ARcW**
>
> >**Q1: Did not significantly improve the performance.**
>
> A: Thank you very much for kindly pointing this out.
> To verify our proposed method, we conduct extensive experiments on popular benchmark datasets and evaluate the robustness against state-of-the-art adversarial attacks, such as AutoAttack[14].
> Evaluation on the AutoAttack is widely adopted [1,2,4,5,6], which is regarded as the most reliable attack algorithm to verify the robustness of the model.   It is very challenging since it takes a wide range of attacks into consideration and reports the worst-case performance.
> Improvement of the 1\% scale is quite hard and significant enough to show the superior performance of the methods [1,2,5,6].
> Under the evaluation budget of 8./255, our method shows superior performance compared with SOTA defending. The results as shown in Table 6 in the appendix.
> Following the suggestions of reviewer Xby6,  we compared with the mentioned work [1,6] under AutoAttack, please kindly refer to the response to Reviewer Xby6.
> The main purpose of our work is to explore the impact of semantic information on the robustness of the model, not just to provide a defense method, although it can achieve great robustness.
>
> >**Q2: Require more resources.**
>
> A: Thank you very much for kindly pointing this out. We do need additional word embedding models to assist training, however, word embedding is one of the critical technologies in natural language processing, there are many third-party open-source libraries that provide a large number of excellent word embedding models (e.g., Glove[15]), which are trained in large-scale training crops. Therefore, we can use existing word embedding models to save training costs.
> We provide the results obtained by training several different word embedding models, as shown in the following table:  We can see that different word embedding models can improve the robustness of the model.
>
> |  Method   | Natural  | AutoAttack  |
> |  ----  | ----  | ----  |
> |    Random Embedding                |  79.69  |  48.87   |
> |    CLIP                            |  80.38  |  50.04   |
> |    Glove (Twitter)                 |  80.71  |  50.12   |
> |     Glove (Common Craw)             |  81.06  |  50.25   |
> |    Glove (Wikipedia 2014 and Gigaword)   |  80.57  |  50.42   |
>
> It would be appreciated if the reviewer can consider upgrading the score if we have provided satisfying answers. Or, it would also be appreciated if the reviewer can let us know whether we could address any parts in more detail. Thank you.

---

### Official Review · Reviewer_FuyA · 2022-10-25

**Confidence:** 4
**Clarity, Quality, Novelty And Reproducibility:** This paper did not provide the code. …
**Correctness:** 3
**Technical Novelty And Significance:** 3
**Empirical Novelty And Significance:** 3
**Recommendation:** 6

**Strength And Weaknesses:**

Strength:
1. The intuition and analysis of the correlation between robust visual representations and semantic word vectors are interesting and meaningful. This opens another lens for understanding the adversarial robustness of vision models and robustness of visual representations.
2. The paper is well-written and the proposed methods are clearly formulated.
3. The theoretical analysis for the lower bound is helpful. But some comparisons between the results w/ and w/o utilizing this estimated lower bound would be appreciated.
Weakness:
1. The similarity analysis in Figure 2 is not well interpreted. How such similarity metrics are normalized is crucial for interpreting the results.
2. The claim in caption of Figure 1 that FGSM is always less robust than standard AT lacks supporting evidences.
3. Comparison with existing methods can be improved. Some of the existing methods can trade off accuracy over robustness (.e.g. TRADES). It might be fairer if the authors also include results of TRADES with different settings.
4. There is a significant performance gap on natural accuracy of models trained with SCARL in CIFAR-10. That is an interesting phenomenon that might be due to limited richness of semantic information in the visual representations. The authors should discuss more on this and other potential limitations.

Some minor things:
P3: nor-robust -> non-robust
Fig 2,3 visible margins of the images. They are also redundant in the appendix.
Other suggestions:
	There is a line of research exploring how semantic word vectors improve the generalization/robustness of the vision models [1] might also be related. Some discussion of differences and future directions would be interesting.

[1] Grounded Language-Image Pre-training


**Summary Of The Paper:**

The authors propose to improve the adversarial robustness of image classification models by adding constraints between visual representations and semantic word vectors. The authors first demonstrate the correlation between visual representations and semantic word vectors and how robust features are aligned with semantic word vectors. Then they propose an adversarial framework (SCARL) to learn robust visual representations through regularizing the structure of visual representations and that of semantic word vectors. As the result, the authors demonstrate the proposed framework improves the model's robustness compared to current approaches.


**Summary Of The Review:**

See above.

---

> ### Author Response · Authors · 2022-11-16
> **Response to Reviewer FuyA**
>
>  >**Q1:The similarity analysis.**
>
> A: Thank you for your question and sorry for the confusion.
> To verify that features of different categories can present analogies and correlations, we visualize the similarity matrix of different features generated by different models.
> In fact, we apply normalized to the features to ensure the vectors are in a unified metric space, then we calculate the similarity (inner product) between different categories of features and get the similarity matrix, as shown in Figure 2.
> The color is brighter with a larger similarity, Therefore, we can see that all features are most similar to themselves.
> For example, in the similarity matrix of the robust features, the correlation between category 3 (Cat) and category 5 (Dog) is stronger than the correlation between category 3 (Cat) and category 9 (Truck).
> The relatedness of the robust features is similar to the relatedness reflected by the semantic word vector.
> We have revised the manuscript to describe this process in detail.
>
>  >**Q2: Comparison with TRADES with different settings.**
>
> A: Thank you very much for the constructive advice. Following the suggestion, we combine the proposed semantic information constraint techniques ($L_{info}$ and $L_{struc}$) with TRADES,
> the balance parameter $\beta$ set as 1 and 6. The results are shown above.
> We can see the robustness can be further improved by combining semantic information (SemInfo).
>
> |  Method   | Natural  |  AutoAttack  |
> |  ----  | ----  | ----  |
> | Adv. training  | 83.14  | 47.56  |
> | Adv. training + SemInfo   | 82.72 | 48.81 |
> | TRADES ($\beta$ = 1)               | 87.49 | 43.80  |
> | TRADES ($\beta$ = 1)  + SemInfo    | 86.88 | 44.34  |
> | TRADES ($\beta$ = 6)               | 82.76 | 49.34  |
> | TRADES ($\beta$ = 6)  + SemInfo    | 82.22 | 49.82  |
>
>  >**Q3: The gap on natural accuracy in CIFAR10, and other limitations.**
>
> Thank you very much for kindly pointing this out. We also noticed this gap. We think there are 2 reasons for this gap:
> 1) The trade-off between clean accuracy and robustness [3,10];
> 2) Under a given dataset, the semantic words of the data category are fixed, when calculating $L_{info}$, even using the large batch size, the number of negative samples in $L_{info}$ is still restricted by the number of semantic words.
> In the CIFAR10 dataset, there are only 10 semantic words, which greatly limits the performance of $L_{info}$.
>
> The other limitations are that some words often have different meanings in different contexts, so they cannot accurately reflect the semantic information of the image. Secondly, there may be multiple objects in the image, causing the visual features of semantic words to fail to match. Currently, our method is suitable for object-center data.
>
>  >**Q4: The FGSM is always less robust than standard AT.**
>
> A: Thank you for your question and sorry for the confusion.
> A large number of experiments[11,12] show that the adversarial training model based on PGD is more robust than that based on FGSM under a strong attack (e.g., Autoattack).
> The possible reason for this fact is FGSM is a weak attack, which can not generate more diverse adversarial examples than PGD attack. Therefore, from the point of view of data augmentation, PGD is a better data augmentation strategy than FGSM since it can generate more diverse samples. In our experiments, the robustness of the FGSM-based model is 39.24 under AutoAttack, while the robustness of the PGD-based model is 47.56.
> Therefore, in our work, we think this assumption can be established.
>
>  >**Q5: About the code**
>
> A:  Thank you very much for pointing this out. We have uploaded the code to reproduce our results in the supplementary material.
>
>  >**Some discussions.**
>
> Thank you very much for the constructive advice. This GLIP [13] presents a grounded language-image pretraining model for learning object-level, language-aware, and semantic-rich visual representations, where detection data introduce more bounding box annotations and help train a multi-grained alignment model.
> Due to the limitations of the adversarial attack, we can only study object-centric images.
> In the future, we will explore whether proposing more better semantic information by extracting more fine-grained text-image pairs can improve the robustness of the model.
> In addition, not only classification tasks, but also object detection and semantic segmentation tasks face the threat of adversarial examples, so we will further explore the impact of semantic information on different tasks.
>
> It would be appreciated if the reviewer can consider upgrading the score if we have provided satisfying answers. Or, it would also be appreciated if the reviewer can let us know whether we could address any parts in more detail. Thank you.

---

### Official Review · Reviewer_Xby6 · 2022-10-30

**Confidence:** 2
**Correctness:** 3
**Technical Novelty And Significance:** 3
**Empirical Novelty And Significance:** 2
**Recommendation:** 5

**Clarity, Quality, Novelty And Reproducibility:**

Applying semantic information in defending against adversarial attacks seems novel.

**Strength And Weaknesses:**

Strengths:
- The motivation for applying semantic information in defending against adversarial attacks seems interesting.

Weaknesses:
- There lacks the SOTA defending models in comparison, such as [A],[B]. The performance gain looks limited compared to the comparison methods (not the latest ones).
- It is not clear how these semantic features come.

[A] “Learnable boundary guided adversarial training.” CVPR 2021.
[B] “Adversarial feature desensitization.” NeurIPS, 2021.


**Summary Of The Paper:**

This paper proposes to utilize semantic information for defending against adversarial attacks. The motivation seems interesting.

**Summary Of The Review:**

The motivation for applying semantic information in defending adversarial attacks seems interesting. However, the effectiveness of the proposed method is not well justified.

---

> ### Author Response · Authors · 2022-11-16
> **Response to Reviewer Xby6**
>
> >**Q1: Compared with several state-of-the-art models.**
>
> A: Thank you very much for the constructive advice. Following the suggestion, we compared with some SOTA defending models, such as LBGAT[1], LAS-AT[2], and so on. The results are shown in the table below. Our SCARL achieves 54.65\% and 56.42\% with AWP respectively, which makes the trained model surpass the previously state-of-the-art models reported by the benchmark (https://robustbench.github.io/). where every small margin of improvement is significant. Notes that our experiments did not use additional datasets. More results are shown in Table 6 in appendix C.
>
> |  Method   |  Architecture  |  Natural   |  AutoAttack  |
> |  ----  | ----  | ----  | ----  |
> | TRADES [3] (ICML2019)             |  WideResNet-34-10 |  84.92 |  53.08  |
> | LBGAT  [1] (ICCV2021)               |  WideResNet-34-20 |  88.70 |  53.57  |
> | Bag of Tricks [4] (ICLR2021)        |  WideResNet-34-20 |  86.43 |  54.39   |
> | LAS-AT [2] (CVPR2022)             |  WideResNet-34-10 |  85.24 |  54.13   |
> | SCARL (Ours)                                |  WideResNet-34-10 |  84.33 |  54.65 |
> | TRADES + AWP  [5] (NeurIPS2020)              |  WideResNet-34-10 |  85.26 |  56.17  |
> | LAS-AT + AWP  [2] (CVPR2022)              |  WideResNet-34-10 |  84.98 |  56.26   |
> | SCARL + AWP                                |  WideResNet-34-10 |  84.70 |  56.42  |
>
> For the AFD [6] as mentioned by reviewer Xby6, which learns adversarially robust features that are both predictive and insensitive to adversarial attacks.
> We checked the results of the AFD, on the CIFAR10 dataset, the robustness of AFD is 59.38\% against PGD attack, while the robustness of AFD is only 37.33\% under AutoAttack.
> Our SCARL can achieve 50.42\% on the CIFAR10 under AutoAttack. All models are baed on ResNet-18.
> Although there are some differences in the experimental settings, it can be seen from the result that AFD cannot defend against AutoAttack [14], which is regarded as the most reliable attack algorithm to verify the robustness of the model.
>
> >**Q2: It is not clear how these semantic features come.**
>
> A: Thank you for your question and sorry for the confusion.
> In order to better explain the semantic features,  we only focus on the essential image classification task.
> A semantic feature needs to meet two requirements: 1) the features must be able to be distinguished; 2) the features must be able to present analogies and correlations between different categories.
> In the process of adversarial training, adversarial attack algorithms tend to look for categories of associative features to attack [7,8,9], so the adversarial examples contain the correlation characteristics between categories, thus the adversarially trained model not only learn the basic feature to be distinguished, but also learn the correlations between different categories. However, normally trained models usually learn the basic features, which only are able to be distinguished. but cannot present analogies and correlations between different categories.
> To verify whether the features contain semantic information, we visualize the similarity matrix of different features generated by different models,
> as shown in figure 2 in the main paper.
> For the feature of the adversarially trained model, the correlation between category 3 (Cat) and category 5 (Dog) is stronger than the correlation between category 3 (Cat) and category 9 (Truck).
> The relatedness of the robust features is similar to the relatedness reflected
> by the semantic word vector.
> However, the features of the normally trained model cannot reflect the association between categories.
> Based on this observation, we introduce semantic word vectors with semantic information, which can better help the model learn semantic features and improve the robustness of models.
>
> It would be appreciated if the reviewer can consider upgrading the score if we have provided satisfying answers. Or, it would also be appreciated if the reviewer can let us know whether we could address any parts in more detail. Thank you.

---

### Author Response · Authors · 2022-11-16
**References**

[1] Learnable boundary guided adversarial training. ICCV, 2021.

[2] LAS-AT: Adversarial Training with Learnable Attack Strategy, CVPR 2022.

[3] Theoretically principled trade-off between robustness and accuracy. ICML, 2019.

[4] Bag of tricks for adversarial training. ICLR, 2021.

[5] Adversarial weight perturbation helps robust generalization. NeurIPS, 2020.

[6] Adversarial feature desensitization. NeurIPS, 2021.

[7] Multi-Targeted Adversarial Example in Evasion Attack on Deep Neural Network. IEEE Access, 2018.

[8] Improving Robust Fairness via Balance Adversarial Training. https://arxiv.org/abs/2209.07534.

[9] An Alternative Surrogate Loss for PGD-based Adversarial Testing. https://arxiv.org/abs/1910.09338.

[10] Robustness May Be at Odds with Accuracy. ICLR, 2019.

[11] Fast is better than free: Revisiting adversarial training. ICLR, 2020.

[12] Bag of Tricks for FGSM Adversarial Training. https://arxiv.org/abs/2209.02684

[13] Grounded Language-Image Pre-training. https://arxiv.org/abs/2112.03857.

[14] Reliable evaluation of adversarial robustness with an ensemble of diverse parameter-free attacks. ICML, 2020.

[15] Glove: Global vectors for word representation. EMNLP, 2014.

---

### Author Response · Authors · 2022-11-16
**Summary of the updates to the revision**

Hi all,

We really thank all reviewers for taking the time to review our paper,
and sincerely appreciate the reviewers' positive feedback on the novelty and significance of our work.
We have updated our manuscript, and the changes are summarized as follows:

(1) We added the results compared with some SOTA models in Appendix C, and more ablation experiments in Appendix F.

(2) We updated the captions of Figure 2. We also fixed some typo errors and updated some images to remove the visible margins.

(3) We uploaded the code to reproduce our results in the supplementary material.

We thank all the reviewers again for their insightful reviews and valuable feedback. We hope that our response has answered the reviewer's question, but please kindly let us know if we misunderstood the question.

The authors.

---

### Author Response · Authors · 2022-12-09
**A kindly Reminder Message**

Dear Reviewers,

We thank all the reviewers again for their insightful reviews and valuable feedback. We hope that our response has answered the reviewer's question, but please kindly let us know If you still have any remaining questions or concerns, we would sincerely like to know and will try our best to resolve them within the remaining discussion period.

Thank you!

The authors

---

### Decision · Program_Chairs · 2023-01-20

**Decision:**

Reject

**Justification For Why Not Higher Score:**

All the reviewers agreed that the paper covers an interesting problem, the use of semantic information to approach adversarial robustness. However, there is a consensus that the paper is not yet ready for publication in its current version.

PC members suggest the following improvement, which may strengthen the paper's future version.
- show stronger performance compared to the state-of-the-art. There could be better ways to explore leveraging semantic information to make the gap clearer or probably use other/additional sources of semantic information.
- experimentation on additional datasets with larger number of classes and hopefully show how the use of semantic information can help as the data/ vocabulary of classes increases.
-Explore how the semantic information may benefit low-resource classes, which has few/no examples

**Justification For Why Not Lower Score:**

 N/A.

**Metareview: Summary, Strengths And Weaknesses:**

The paper introduces an approach for adversarial robustness using semantic information to assist model training. Specifically, the authors propose Semantic Constraint Adversarial Robust Learning (SCARL) and demonstrate the impact of their approach on model robustness on multiple benchmarks.

Strengths
--------------

-The idea of applying semantic information in defending against adversarial attacks is well-motivated and interesting (Reviewer Xby6,  Reviewer FuyA )
-The paper is well-written  and organized ( Reviewer FuyA and Reviewer ARcW)
-Sound theoretical analysis ( Reviewer FuyA)
- The performance is generally competitive compared to prior arts (Reviewer ARcW)


Weaknesses
---------------
-The performance gain compared to SOTA is limited; small margins (all reviewers)
- Experiments with TRADE were requested by Reviewer FuyA, but the improvement is also not very large.
- Additional minor issues with Fig2 and missing discussions that were partially addressed in the author response period.


The reviewers recommended
-5: marginally below the acceptance threshold
-6: marginally above the acceptance threshold
-5: marginally below the acceptance threshold


**Summary Of Ac-Reviewer Meeting:**

As a borderline paper, we had an AC-reviewer meeting, and the meeting concluded that the paper indeed has merit. However, the biggest concern that all reviewers agreed on is that the improvements over previous methods are not convincing enough (small margins), especially considering that the proposed method requires more information, like pre-trained word embedding.